# Connecting the Dots: Generating Realistic Tabular Data with Structural Causal Models

## Abstract

Most tabular-data generators match marginal statistics yet ignore causal structure, leading downstream models to learn spurious or unfair patterns. We present Tab-SCM, a mixed-type generator that preserves those causal dependencies. Starting from a Completed Partially Directed Acyclic Graph (CPDAG) found by any discovery algorithm, TabSCM (i) orients edges to a DAG, (ii) fits root-node marginals with KDE or categorical frequencies, and (iii) learns topologically ordered structural assignments: conditional diffusion models for continuous children and gradient-boosted trees for categorical ones. Ancestral sampling yields semantically valid records and enables exact counterfactual queries. On seven public datasets, encompassing healthcare, finance, housing, environment, Tab-SCM matches or surpasses state-of-the-art GAN, diffusion, and LLM baselines in statistical fidelity, downstream utility, and privacy risk, while also cutting rule-violation rates and providing causally meaningful and robust counterfactual interventions. Because generation is decomposed into explicit equations, it runs up to $583\times$ faster than diffusion-only models and exposes interpretable knobs for fairness auditing and policy simulation, making TabSCM a practical choice for realism, explainability, and causal soundness.

## 1 Introduction

Synthetic data is increasingly recognized as a practical solution to many of the challenges associated with real-world data: privacy constraints Jordon et al. (2018); Chen et al. (2021), data sparsity Esteban et al. (2017); Frid-Adar et al. (2018), access restrictions Goncalves et al. (2020), and fairness concerns Veale & Binns (2017); Xu et al. (2018); Barbierato et al. (2022). In regulated domains such as healthcare, finance, and education, where the availability of labeled, high-quality data is often limited, synthetic data can offer a privacy-preserving and compliance-friendly alternative to real data Goncalves et al. (2020); Jordon et al. (2018).

While substantial progress has been made in generating realistic synthetic data for images, text, and time series Radford et al. (2018); Ramesh et al. (2022); Dhariwal & Nichol (2021); Rombach et al. (2022), tabular data remains a uniquely challenging modality. Tabular datasets frequently encode heterogeneous data types, non-linear dependencies, and causal relationships among variables. These causal de-

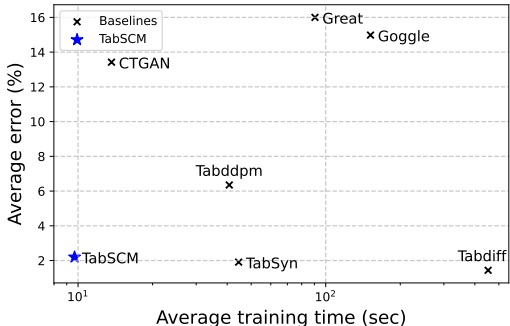

Figure 1: Shows average error (Eq. 3) and average training time of TabSCM against each baseline method averaged over seven real-world datasets.

pendencies are essential for supporting counterfactual reasoning, robust decision-making, and fair model behavior in high-stakes applications such as credit risk assessment, treatment effect estimation, and resource allocation Louizos et al. (2017); Johansson et al. (2016).

Despite recent advances in generative modeling for tabular data, including methods such as TabD-DPM Kotelnikov et al. (2023), TabSyn Zhang et al. (2024), TabDiff Shi et al. (2025), and GReaT Borisov et al. (2023), the evaluation of generated data largely focuses on statistical fidelity (e.g., marginal/conditional distributions, pairwise correlations) or utility-based metrics (e.g., downstream task performance). These criteria, however, are agnostic to the preservation of causal structure, which is fundamental for ensuring that synthetic data supports valid inferences and avoids reinforcing spurious patterns.

Moreover, existing state-of-the-art models often struggle to generate semantically valid samples but generate out-of-domain entries that violate known constraints, see Table 4. Such issues limit their applicability in real-world decision-making and high-stakes environments for which validity matters.

To address these limitations, we propose TabSCM, a synthetic data generation framework that leverages structural causal models (SCMs) to explicitly model and preserve the underlying causal relationships in the data. By conditioning on the causal graph, TabSCM generates samples in a topologically ordered fashion, ensuring semantic validity, supporting counterfactual reasoning, and offering greater transparency. In contrast to diffusion- or transformer-based approaches, our method is both significantly faster and more interpretable, while producing higher-quality and causally coherent synthetic data. This makes TabSCM a practical and principled tool for fair, explainable, and policy-aware data generation.

In summary, our contributions are:

(1) **A practical framework for mixed tabular data**: We propose TabSCM, leveraging Structural Causal Models (SCMs) that combine causal reasoning with decision trees and diffusion models. By leveraging the causal structure, our method improves the computational time by up to $583\times$ in comparison to state-of-the-art diffusion methods. TabSCMs modularity significantly decreases computational time on average while providing state-of-the-art statistical fidelity, see Section 1.

(2) **Realistic, valid, and privacy-preserving synthetic data**: Through extensive experiments, see Section 6, we prove that our proposed method achieves competitive or better results in statistical fidelity, utility, and privacy of generated data, and excels in the validity of generated samples in contrast to diffusion-only methods.

(3) **Additional insights with counterfactual interventions**: We show that our method comes with a natural extension of counterfactual interventions. This property is a useful addition and enables meaningful simulations of "what-if" scenarios and enables the generation of out-of-distribution (OOD) samples.

Our evaluation shows that TabSCM is able to generate privacy-preserving and valid samples while maintaining a high utility for downstream learning tasks.

## 2 RELATED WORK

**Causal aware generation.** Causal aware generation integrates knowledge of cause-and-effect relationships between variables into the synthetic data generation process. Unlike traditional generative models that rely solely on statistical correlations, causal models generate data in a way that respects the underlying structural dependencies. Causal-GAN Kocaoglu et al. (2018) was one of the first to incorporate causality into a Generative Adversarial Network (GAN). Causal-TGAN Wen et al. (2022) is an adaptation tailored to tabular data. While delivering promising results, the evaluation is limited, putting the generalizability in question. GOGGLE Liu et al. (2023) is another framework that integrates causal graphs and generative modeling for tabular data generation. It directly learns the adjacency matrix and leverages the causal structure, generating realistic samples. Both methods deploy GANs. Contrarily, we use diffusion-based models to increase the sample quality in error rates, downstream utility, and privacy (see Section 6).

**Generative models for tabular data generation.** Over the last years, Generative Adversarial Networks (GANs), Diffusion models, and LLMs have emerged as popular frameworks for tabular data generation. While GAN-based methods TableGAN Park et al. (2018), CTGAN Xu et al. (2019), and TVAE Xu et al. (2019) paved the way for deep learning for tabular data synthesis, they often fail to capture the full diversity of the real data distribution. With the rise of diffusion-based architectures in a wide range of deep learning applications, many diffusion-based tabular data generators such

as TabSyn Zhang et al. (2024), TabDDPM Kotelnikov et al. (2023), and TabDiff Shi et al. (2025) showed promising results in tabular data synthesis. The recent success of extracting and generating language with LLMs led to various LLM-based methods for tabular data synthesis. One early work includes GReaT Borisov et al. (2023) encoding tabular data into meaningful tokens, which are then used to finetune an LLM. Various methods, including Pred-LLM Nguyen et al. (2024), Tabula Zhao et al. (2023), Tabby Cromp et al. (2024), followed the approach and exploited LLMs to a certain extent for generating tabular data.

## 3 PROBLEM SETUP

Let $\mathcal{D}_{\text{real}} \in \mathbb{R}^{n \times d}$ denote a real-world tabular dataset consisting of $n$ samples and $d$ variables. Our objective is to evaluate how well synthetic data generators preserve the causal structure encoded in $\mathcal{D}_{\text{real}}$. We represent the causal structure of the data using a causal graph:

$$\mathcal{G} = (\mathcal{V}, \mathcal{E}),$$

where $\mathcal{V} = \{X_1, X_2, \ldots, X_d\}$ is the set of observed variables (nodes), and $\mathcal{E} \subseteq \mathcal{V} \times \mathcal{V}$ is the set of directed edges representing direct causal relationships between variables, i.e., $(X_i \to X_j) \in \mathcal{E}$ implies that $X_i$ is a direct cause of $X_j$. For our theoretical framework, we assume the causal sufficiency condition holds (i.e., no unobserved confounders), and that the data-generating process is Markovian and faithful to a Directed Acyclic Graph (DAG) denoted by $\mathcal{G}$.

A *v-structure* is a triplet of nodes $(X_i, X_j, X_k)$ such that $(X_i \to X_k)$ and $(X_j \to X_k)$, where the nodes $X_i, X_j$ are not adjacent (i.e., not connected by and edge). A *Markov equivalence class* (MEC) consists of DAGs encoding the same set of conditional independence relations among the nodes.

Figure 2: Minimal example of a system of four observed variables $X_i$, and corresponding exogenous variables $\epsilon_i$ for $i = 1, 2, 3, 4$. The causal relationships and interactions of the observed variables are illustrated on the left-hand side (causal graph $\mathcal{G}$). On the right-hand side, we describe the SCM for the associated causal graph $\mathcal{G}$ on the left.

A *Structural Causal Model* (SCM) provides a formal framework for representing and reasoning about cause-effect relationships between variables in a system. Figure 2 illustrates a minimal example of an SCM. Formally, an SCM is defined as a tuple

$$\mathcal{M} = (\mathcal{G}, \mathcal{F}, E, \mathbb{P}_E),$$

where $\mathcal{G} = (\mathcal{V}, \mathcal{E})$ denotes a causal graph, $E = (\epsilon_1, \ldots, \epsilon_d)$ is a set of exogenous variables (noise), $\mathcal{F}$ is a collection of structural assignments (or structural equations), where each $X_i$ is defined as

$$X_i := f_i(\mathbf{PA}_i, \epsilon_i), \tag{1}$$

where $\mathbf{PA}_i$ denotes the set of parent variables of $X_i$ in $\mathcal{G}$ Pearl (2000); Peters et al. (2017). The joint distribution of the exogenous variables $\mathbb{P}_E$ are mutually independent such that

$$\mathbb{P}_E := \prod_{i=1}^{d} \mathbb{P}(\epsilon_i).$$

The graph structure and the mutual independence of the exogenous variables enable the causal factorization of the joint probability distribution of the observables

$$\mathbb{P}(X_1, \ldots, X_d) = \prod_{i=1}^{d} \mathbb{P}(X_i | \mathbf{PA}_i), \tag{2}$$

disentangling it into conditionals according to the structural assignments Pearl (2000).

## 4 PROPOSED METHOD

Here, we describe TabSCM, a model for tabular data generation utilizing a structural causal model (SCM).

Consider the topological ordering of the structural assignments (1) linked to each node of the underlying graph

$$f_{\pi^{-1}(1)} \prec f_{\pi^{-1}(2)} \prec \cdots \prec f_{\pi^{-1}(d)},$$

where $\pi : \{1, \ldots, d\} \to \{1, \ldots, d\}$ denotes the permutation of the set of edges such that $f_{\pi^{-1}(1)} \prec f_{\pi^{-1}(2)}$ indicates that node $\pi^{-1}(1)$ is a parent of node $\pi^{-1}(2)$. Our proposed method follows the topological ordering and starts at the root nodes. We illustrate the conceptual framework of TabSCM in Figure 3.

**Root nodes.** For each root node $\{X_i : \mathbf{PA}_i = \emptyset; i = 1, \ldots, d\} \in \mathcal{V}$, we directly estimate its marginal distribution $\mathbb{P}(X_i)$ based on the observed data $\{x_j^i\}_{j=1}^n$ from $\mathcal{D}_{\text{real}}$. For continuous variables, we apply kernel density estimation (KDE)

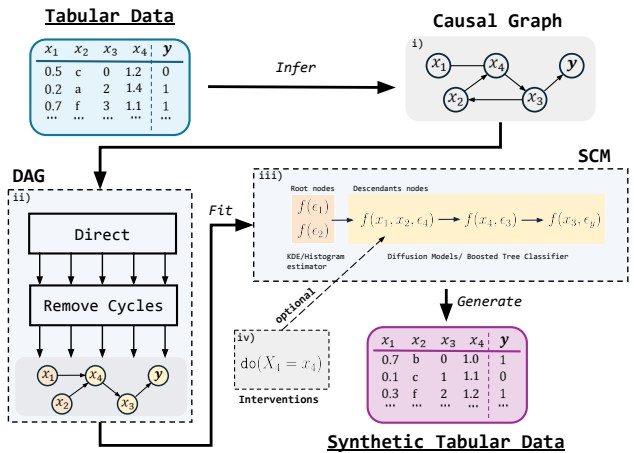

Figure 3: The conceptual framework of our proposed method, including i) causal discovery, ii) refining inferred Causal graph, iii) learning structural assignments and conditional sampling, and iv) Counterfactual Interventions.

$$\hat{\mathbb{P}}(x) = \frac{1}{nh} \sum_{j=1}^n K\left(\frac{x - x_j}{h}\right),$$

with a Gaussian kernel function $K(u) := \frac{1}{\sqrt{2\pi}} \exp\left(\frac{-u^2}{2}\right)$. For categorical root nodes $X_i$, we estimate the marginal distribution with relative frequencies for $c = 1, \ldots, C$ with:

$$\hat{\mathbb{P}}(X_i = c) = \frac{1}{n} \sum_{j=1}^n \mathbf{1}[x_j^{(i)} = c].$$

**Conditional nodes.** For each continuous non-root node $X_i \in \mathcal{V}$ with parent set $\mathbf{PA}_i \neq \emptyset$, we aim to model the conditional distribution $\mathbb{P}(X_i \mid \mathbf{PA}_i)$ using a conditional denoising diffusion probabilistic model (DDPM). The diffusion model defines a forward process (diffusion) that gradually adds Gaussian noise to the target variable $X_i$, and a reverse process that learns to denoise and reconstruct $X_i$ given its parents. Given a normalized input $x_i^{(0)} \sim \mathbb{P}(X_i \mid \mathbf{PA}_i)$, the forward process corrupts it by gradually adding Gaussian noise $q(x_i^{(t)} \mid x_i^{(0)}) = \mathcal{N}(x_i^{(t)}; \sqrt{\bar{\alpha}_t} x_i^{(0)}, (1 - \bar{\alpha}_t)\mathbf{I})$, where $x_i^{(0)}$ is the clean data sample, $x_i^{(t)}$ is the noisy version at timestep $t$, and $\{\bar{\alpha}_t\}_{t=1}^T$ is the cumulative product of noise schedule coefficients

$$\bar{\alpha}_t = \prod_{s=1}^t \alpha_s, \quad \alpha_s = 1 - \beta_s.$$

We train a neural network $h_\theta$ to predict the noise added at each step, conditioned on the parents $\mathbf{PA}_i$ and the uniformly distributed diffusion timestep $t$,

$$\hat{h} = h_\theta(x_i^{(t)}, \mathbf{PA}_i, t).$$

We train the model to minimize the mean-squared error between true and predicted noise,

$$\mathcal{L}_t = \mathbb{E}_{(x_i^{(0)}, \epsilon, t)} \left[ \left\| \epsilon - h_\theta\left(x_i^{(t)}, \mathbf{PA}_i, t\right) \right\|_2^2 \right].$$

For categorical nodes, we deploy a boosted tree classifier Friedman (2001) providing a flexible and robust approximation.

**Counterfactual reasoning.** Structural Causal Model enable counterfactual reasoning, which is the ability to ask what would have happened if a variable had taken a different value. This is achieved by explicitly modeling how variables in a system interact through causal mechanisms (i.e., functional assignments that describe how each variable is generated from its causes and some noise). The key

idea is that by altering these mechanisms only for specific variables, we can simulate alternative, hypothetical *counterfactual* instances. Formally, a counterfactual intervention refers to modifying the data-generating process by setting a variable $X_i \in \mathcal{V}$ to a fixed value $x_i$, and keeping the rest of the model unchanged. This is denoted by the do-operator $\mathrm{do}(X_i = x_i)$, following the framework introduced by Pearl (2000; 2012). The intervention replaces the structural assignment for $X_i$ with a fixed assignment

$$X_i = x_i,$$

resulting in a new SCM denoted by $\mathcal{M}_{X_i = x_i}$. Following the topological ordering, all descendant variables are generated based on the fixed assignment $X_i = x_i$ instead of the original functional assignment $f_i(\mathbf{PA}_i, \epsilon_i)$. In practice, sampling from the joint distribution w.r.t. the intervention $X_j = x_j$ includes:

**(1) SAMPLE EXOGENOUS VARIABLES:** Draw sample of the noise vector $E = (\epsilon_1, \ldots, \epsilon_d) \sim \mathbb{P}_E$, with i.i.d. $\epsilon_i$.

**(2) SET INTERVENTION:** Replace the structural assignment for $X_j$ with $X_j := x_j$, removing the dependence on $\mathbf{PA}_j$ and $\epsilon_j$. This step replaces the causal mechanism for $X_j$ and is the main difference to conditioning which leaves the structural assignment intact.

**(3) FORWARD SAMPLING:** Follow the topological ordering and generate $x_i$ accordingly

$$x_i = \begin{cases} x_j & \text{if } i = j \\ x_i \sim \mathbb{P}(X_i | \mathbf{PA}_i) & \text{otherwise} \end{cases}$$

Steps (1)–(3) generate samples from the **interventional distribution**

$$\mathbb{P}\big(\mathbf{X} \mid \mathrm{do}(X_j = x_j)\big).$$

For **unit-level counterfactuals**, one would first condition on the factual observation to infer the posterior exogenous noise $\hat{E} \sim \mathbb{P}\big(E \mid \mathbf{X} = \mathbf{x}_{\mathrm{obs}}\big)$, and then reuse the same $\hat{E}$ in Steps (2)–(3).

## 5 EXPERIMENTAL SETUP

### 5.1 DATASETS

We use a total of seven real-world datasets covering classification and regression tasks from various application domains. The data covers large-scale datasets ($>$250k samples) and small-scale datasets ($<$1000 samples).

**Classification.** The **Adult** Census Income Becker & Kohavi (1996) dataset contains demographic and employment-related information from the 1994 U.S. Census to predict whether an individual earns more than \$50,000 annually. The Early Stage **Diabetes** Risk Prediction Islam et al. (2019) dataset contains 520 patient records collected via questionnaires at Sylhet Diabetes Hospital in Bangladesh, including 16 demographic and symptom-related features such as age, gender, polyuria, polydipsia, and sudden weight loss. The task is to predict whether an individual is diabetic. The Home Equity Line of Credit (**HELOC**) dataset[1] contains anonymized credit report features with 24 numeric variables detailing borrowers' credit behaviors. The task is to classify whether an applicant will default (or fail to repay) their HELOC within two years. The **Loan** dataset[2] contains over 250k historic consumer loan applications from India, including demographic, financial, and behavioral attributes. The task is to classify applicants into likely defaulters or reliable borrowers. The **Magic** Gamma Telescope dataset Bock (2004) simulates the detection of high-energy gamma particles using a ground-based atmospheric Cherenkov telescope and imaging techniques. The task is to classify events as gamma-ray signals or background cosmic-ray-induced hadronic showers.

**Regression.** The **Beijing** PM2.5 dataset Liang et al. (2015) includes hourly PM2.5 measurements from the U.S. Embassy in Beijing, along with meteorological data from Beijing Capital International Airport. The goal is to predict PM2.5 concentration levels. The California **Housing** dataset Pace & Barry (1997) contains district-level demographic and housing information across California. The task is to predict the median house value for each district.

---

[1]https://tinyurl.com/2r4mxjbp
[2]https://tinyurl.com/44eex2yp

## 5.2 EVALUATION METRICS

Here, we briefly describe the metrics used for each evaluation category along three dimensions i) statistical similarity, ii) downstream utility, and iii) privacy.

**Statistical Similarity:** We follow the two-folded evaluation of the statistical similarity proposed by SDMetrics[3] and used by prior works Zhang et al. (2024); Shi et al. (2025). This includes column-wise density estimation and correlation error calculation.

For each continuous column $i \in C_{\text{num}}$, we use the Kolmogorov–Smirnov (KS) test, which quantifies the maximum absolute difference between the empirical cumulative distribution functions (CDFs) of the real and synthetic data. The KS statistic is defined as

$$\text{KS}(i) := \sup_x \left| F_i^{\text{real}}(x) - F_i^{\text{syn}}(x) \right|,$$

where $F_i^{\text{real}}, F_i^{\text{syn}}$ denote the CDFs of the $i$-th column of the real and synthetic data. Lower values indicate closer alignment between the two distributions. For categorical columns $i \in C_{\text{cat}} := C \setminus C_{\text{num}}$, we compute the Total Variation (TV) distance between empirical distributions $p_i^{\text{real}}$ and $p_i^{\text{syn}}$:

$$\text{TV}(i) := \frac{1}{2} \sum_{a \in \mathcal{A}_i} \left| p_i^{\text{real}}(a) - p_i^{\text{syn}}(a) \right|,$$

where $\mathcal{A}_i$ is the set of categories for column $i$. We report the average distance

$$\mathbf{e}_{\text{den}} := \frac{1}{|C|} \left( \sum_{i \in C_{\text{num}}} \text{KS(i)} + \sum_{i \in C_{\text{cat}}} \text{TV(i)} \right). \tag{3}$$

To assess how well the synthetic data preserves pairwise dependencies, we compute separate errors for numerical and categorical pairs, and aggregate them into a correlation error score. We denote the set of all possible combinations of numerical columns $\mathcal{I}_{\text{num}} := \{(i,j) \in C_{\text{num}} \times C_{\text{num}}\}$ for $|C_{\text{num}}| \geq 1$. For numerical column pairs $(i,j)$, we compute the Pearson correlation coefficients $\rho_{ij}^{\text{real}}$ and $\rho_{ij}^{\text{syn}}$, and define the numerical correlation error as

$$\mathbf{e}_{\text{corr}}^{\text{num}} := \frac{1}{|\mathcal{I}_{\text{num}}|} \sum_{(i,j) \in \mathcal{I}_{\text{num}}} \left| \rho_{ij}^{\text{real}} - \rho_{ij}^{\text{syn}} \right|.$$

For categorical column pairs $(i,j)$, we construct empirical contingency tables $R^{(ij)}$ and $S^{(ij)}$ from the real and synthetic datasets, respectively. The categorical correlation error is defined as the Total Variation distance between the normalized contingency tables:

$$\mathbf{e}_{\text{corr}}^{\text{cat}} := \frac{1}{|\mathcal{I}| - |\mathcal{I}_{\text{num}}|} \sum_{(i,j)} \frac{1}{2} \sum_{\alpha,\beta} \left| R_{\alpha,\beta}^{(ij)} - S_{\alpha,\beta}^{(ij)} \right|.$$

For mixed pairs $(i,j)$ with one numerical and one categorical column, we discretize the numerical variable into bins and apply the same procedure as for categorical pairs. Finally, we define the overall correlation error as the average of the numerical and categorical components

$$\mathbf{e}_{\text{corr}} := \frac{1}{2} \left( \mathbf{e}_{\text{corr}}^{\text{num}} + \mathbf{e}_{\text{corr}}^{\text{cat}} \right). \tag{4}$$

Lower values of $\mathbf{e}_{\text{corr}}$ indicate that the synthetic data preserves the correlation structure of the real data more accurately.

**Utility:** Following prior work Zhang et al. (2024); Shi et al. (2025); Liu et al. (2023), we assess the utility of synthetic data by training an XGBoost classifier or regressor on synthetic samples and evaluating performance on the real test set. Specifically, we split each real dataset into training and test sets, train the generative model on the real training data, and generate a synthetic dataset of equal size. An XGBoost classifier/regressor Chen & Guestrin (2016) is trained on this synthetic data using hyperparameters selected via grid search on 20 random train/validation splits, and evaluated on the real test set. We report the mean and standard deviation of AUC (for classification tasks) or RMSE (for regression tasks) across these runs.

---

[3]https://docs.sdv.dev/sdmetrics

| Method | Adult | Beijing | Diabetes | HELOC | Housing | Loan | Magic |
|---|---|---|---|---|---|---|---|
| GReaT | $56.6 \pm 0.03$ | $7.21 \pm 0.08$ | $5.46 \pm 0.85$ | $12.56 \pm 0.11$ | $9.19 \pm 0.09$ | $4.05 \pm 0.01$ | $16.9 \pm 0.24$ |
| CTGAN | $17.4 \pm 0.04$ | $18.6 \pm 0.08$ | $7.31 \pm 0.34$ | $18.1 \pm 0.07$ | $8.79 \pm 0.04$ | $13.1 \pm 0.02$ | $10.3 \pm 0.09$ |
| TabDDPM | $1.10 \pm 0.07$ | $\underline{1.22 \pm 0.05}$ | $4.63 \pm 0.26$ | $2.79 \pm 0.13$ | $44.7 \pm 0.05$ | $31.1 \pm 0.06$ | $1.17 \pm 0.26$ |
| TabSyn | $\underline{0.84 \pm 0.07}$ | $1.29 \pm 0.04$ | $2.60 \pm 0.39$ | $\mathbf{2.09 \pm 0.13}$ | $1.17 \pm 0.13$ | $4.59 \pm 0.02$ | $\underline{1.15 \pm 0.12}$ |
| TabDiff | $\mathbf{0.75 \pm 0.02}$ | $\mathbf{1.16 \pm 0.07}$ | $\underline{2.48 \pm 0.73}$ | $2.27 \pm 0.09$ | $\underline{1.56 \pm 0.16}$ | $1.34 \pm 0.01$ | $\mathbf{0.79 \pm 0.11}$ |
| GOGGLE | $13.9 \pm 0.05$ | $20.1 \pm 0.08$ | $37.1 \pm 0.17$ | $4.55 \pm 0.08$ | $10.9 \pm 0.08$ | OOM | $3.35 \pm 0.09$ |
| TabSCM (Ours) | $2.46 \pm 0.09$ | $1.92 \pm 0.05$ | $\mathbf{1.73 \pm 0.16}$ | $\mathbf{2.09 \pm 0.08}$ | $2.36 \pm 0.05$ | $\mathbf{1.21 \pm 0.02}$ | $3.90 \pm 0.05$ |

Table 1: The error rate (%) of column-wise density estimation (Eq. 3). Lower values indicate a more accurate estimation (superior results). **Bold** numbers indicate best performance; underlined values are second best results for each dataset.

| Method | Adult | Beijing | Diabetes | HELOC | Housing | Loan | Magic |
|---|---|---|---|---|---|---|---|
| GReaT | $80.9 \pm 0.09$ | $9.82 \pm 3.20$ | $10.5 \pm 0.93$ | $8.29 \pm 0.15$ | $15.9 \pm 1.73$ | $22.06 \pm 0.03$ | $10.5 \pm 0.39$ |
| CTGAN | $18.3 \pm 0.89$ | $20.3 \pm 0.03$ | $13.8 \pm 0.38$ | $7.64 \pm 0.07$ | $16.4 \pm 0.24$ | $27.0 \pm 0.03$ | $11.1 \pm 0.17$ |
| TabDDPM | $2.11 \pm 0.09$ | $4.45 \pm 0.12$ | $20.5 \pm 0.15$ | $1.99 \pm 0.21$ | $21.5 \pm 0.04$ | $55.0 \pm 0.07$ | $1.20 \pm 0.41$ |
| TabSyn | $\underline{1.94 \pm 0.44}$ | $\underline{3.48 \pm 0.26}$ | $\underline{4.65 \pm 0.82}$ | $\mathbf{1.71 \pm 0.33}$ | $\underline{1.89 \pm 0.21}$ | $11.7 \pm 0.03$ | $\mathbf{0.75 \pm 0.09}$ |
| TabDiff | $\mathbf{1.59 \pm 0.02}$ | $2.94 \pm 0.14$ | $\mathbf{3.95 \pm 0.33}$ | $1.93 \pm 0.17$ | $3.0 \pm 0.03$ | $\underline{8.83 \pm 0.02}$ | $\underline{0.81 \pm 0.13}$ |
| GOGGLE | $25.1 \pm 0.09$ | $46.6 \pm 0.05$ | $46.9 \pm 0.24$ | $10.8 \pm 0.26$ | $22.3 \pm 0.20$ | OOM | $9.35 \pm 0.53$ |
| TabSCM (Ours) | $5.12 \pm 0.11$ | $3.89 \pm 0.05$ | $6.75 \pm 0.17$ | $\underline{1.86 \pm 0.48}$ | $\mathbf{1.86 \pm 0.04}$ | $\mathbf{6.62 \pm 0.06}$ | $2.79 \pm 0.53$ |

Table 2: The error rate (%) (Eq. 4) of correlation estimation between column distribution of real and synthetic data. **Bold** numbers indicate best performance; underlined values are second best results for each dataset.

**Privacy:** Distance to Closest Records (DCR) is a privacy-related metric used to assess how similar synthetic data points are to real data points. For $x \in \mathcal{D}_{\text{Syn}}$, the DCR is

$$\text{DCR}(x) = \min_{x_j \in \mathcal{D}_{\text{real}}} ||x - x_j||_1,$$

and quantifies the distance to the nearest real record. A small value (DCR$\approx 0$) leaks real information, a moderate value is associated with low privacy risk, and a high DCR value is safe from a privacy perspective but might have low utility in general Park et al. (2018). Our evaluation shows that TabSCM is able to generate privacy-preserving samples while maintaining a high utility for downstream learning tasks.

## 6 EXPERIMENTAL RESULTS

We conducted all experiments on an NVIDIA RTX 6000 Ada 48GB. For GReaT, TabDDPM, and TabSyn, we used the hyperparameters described in Zhang et al. (2024). For GOGGLE, we set the dimension of the encoder to $512$, that of the decoder to $128$, and replaced the fixed random seed in the sampling phase. For TabDiff, we used the hyperparameters described in Shi et al. (2025). For CTGAN, we also used the default parameter set. Whenever possible, we report the mean and std deviation for each metric obtained from the evaluation over five trials. The choice of causal discovery method and hyperparameter setting for TabSCM is described in Section A.5.3 and summarized in Table 18.

**TabSCM achieves SoTA performance, matching or surpassing full diffusion models while maintaining a low approximation error of the marginal distributions**. Notably, TabSCM significantly outperforms deep generative models in 3 out of 7 datasets, delivering a stable and generalizable behavior, see Table 1. GOGGLE could not be applied to the large-scale Loan dataset due to memory issues, and TabDDPM was not able to generate meaningful samples for the Housing dataset.

While the error of the column-wise density estimation is a valid indicator of whether the generative model is able to learn the distribution for each feature individually, the feature correlation is a key indicator of whether the generated data behaves realistically. TabSCM is able to outperform all baseline methods in 2 out of 7 datasets, notably this included the large-scale dataset loan, see

Table 2. The derived causal graph models the relationship of each variable, therefore the quality of the correlation error of TabSCM is linked to the causal discovery algorithm. TabSCM factorizes the data-generating process into modular components, abolishing this modularity, e.g., TabSyn and TabDiff, can lead to lower correlation errors on average.

**TabSCM achieves SoTA performance, matching or surpassing full diffusion models, while enhancing privacy and providing high downstream utility**. Generally, a low error of the column-wise density and correlation error implies high downstream utility, see Tables 1 to 3. TabSCM delivers competitive downstream utility and even outperforms the diffusion-only models, e.g., Tab-Diff, TabSyn, and TabDDPM, which overall excel in this metric, on two datasets. While downstream utility is a valid indicator of whether generated samples may be used to replace or augment real data, it can be achieved by copying the data, which violates privacy. TabSCM delivers a strong performance considering the trade-off between utility and privacy and shows an average relative deviation of 5% from the best AUC/RMSE result while increasing the DCR by 0.74 on average.

| Method | Adult | | Beijing | | Diabetes | | HELOC | |
|---|---|---|---|---|---|---|---|---|
| | AUC ($\uparrow$) | DCR | RMSE ($\downarrow$) | DCR | AUC ($\uparrow$) | DCR | AUC ($\uparrow$) | DCR |
| Real | **0.927** | - | **0.431** | - | **0.975** | - | **0.814** | - |
| GReaT | $0.768 \pm .171$ | $7.036 \pm 2.18$ | $0.754 \pm .028$ | $1.579 \pm 0.02$ | $0.687 \pm .113$ | $2.26 \pm 0.11$ | $0.795 \pm .006$ | $0.489 \pm 0.01$ |
| CTGAN | $0.885 \pm .004$ | $1.389 \pm 0.02$ | $0.854 \pm .022$ | $1.749 \pm 0.01$ | $0.544 \pm .009$ | $4.94 \pm 0.03$ | $0.756 \pm .005$ | $0.925 \pm 0.02$ |
| TabDDPM | $0.908 \pm .002$ | $0.589 \pm 0.02$ | $\underline{0.585 \pm .005}$ | $1.439 \pm 0.01$ | $0.368 \pm .146$ | $5.51 \pm 0.05$ | $\underline{0.808 \pm .003}$ | $0.603 \pm 0.01$ |
| TabSyn | $\underline{0.909 \pm .001}$ | $0.679 \pm 0.01$ | $\mathbf{0.568 \pm .012}$ | $1.525 \pm 0.01$ | $\mathbf{0.991 \pm .007}$ | $1.13 \pm 0.07$ | $0.789 \pm .009$ | $0.639 \pm 0.01$ |
| TabDiff | $\mathbf{0.912 \pm .002}$ | $0.555 \pm 0.01$ | $\mathbf{0.568 \pm .013}$ | $1.351 \pm 0.01$ | $\underline{0.982 \pm .011}$ | $1.03 \pm 0.06$ | $0.798 \pm .005$ | $0.689 \pm 0.01$ |
| GOGGLE | $0.814 \pm .009$ | $1.105 \pm 0.01$ | $1.226 \pm .013$ | $1.855 \pm 0.02$ | $0.632 \pm .009$ | $7.42 \pm 0.02$ | $0.378 \pm .004$ | $0.892 \pm 0.01$ |
| TabSCM (Ours) | $0.842 \pm .008$ | $1.554 \pm 0.01$ | $0.594 \pm .014$ | $1.729 \pm 0.01$ | $0.945 \pm .028$ | $3.40 \pm 0.04$ | $\mathbf{0.813 \pm .004}$ | $0.685 \pm 0.01$ |

| Method | Housing | | Loan | | Magic | |
|---|---|---|---|---|---|---|
| | RMSE ($\downarrow$) | DCR | AUC ($\uparrow$) | DCR | AUC ($\uparrow$) | DCR |
| Real | **0.188** | - | **0.921** | - | **0.948** | - |
| GReaT | $0.265 \pm .007$ | $0.084 \pm 0.01$ | $0.530 \pm .002$ | $3.136 \pm 0.02$ | $0.881 \pm .003$ | $0.157 \pm 0.01$ |
| CTGAN | $0.352 \pm .012$ | $0.158 \pm 0.01$ | $0.490 \pm .001$ | $5.183 \pm 0.02$ | $0.821 \pm .006$ | $0.362 \pm 0.01$ |
| TabDDPM | $0.594 \pm .07$ | $2.955 \pm 0.01$ | $0.507 \pm .009$ | $5.833 \pm 0.01$ | $0.932 \pm .002$ | $0.199 \pm 0.01$ |
| TabSyn | $\mathbf{0.235 \pm .004}$ | $0.109 \pm 0.01$ | $\underline{0.561 \pm .001}$ | $4.771 \pm 0.01$ | $\underline{0.934 \pm .005}$ | $0.199 \pm 0.01$ |
| TabDiff | $\underline{0.241 \pm .011}$ | $0.118 \pm 0.01$ | $0.506 \pm .001$ | $5.423 \pm 0.01$ | $\mathbf{0.936 \pm .006}$ | $0.202 \pm 0.01$ |
| GOGGLE | $0.381 \pm .003$ | $0.264 \pm 0.01$ | OOM | OOM | $0.876 \pm .002$ | $0.331 \pm 0.01$ |
| TabSCM (Ours) | $0.253 \pm .005$ | $0.114 \pm 0.01$ | $\mathbf{0.591 \pm .010}$ | $3.556 \pm 0.01$ | $0.928 \pm .001$ | $0.256 \pm 0.01$ |

Table 3: Accuracy of classifiers/regressors trained on synthetic data evaluated on real test data. **Bold** numbers indicate best performance; underlined values are second best results for each dataset. DCR values are not highlighted.

**TabSCM demonstrates low violation rates outperforming full diffusion models on 3 out of 4 sanity checks**. We conduct sanity checks of the Adult, Housing, and Loan dataset. We investigate if longitude and latitude imply a location inside of California (S1), education implies the same education number as given by the real data (S2), if husband or wife implies the appropriate gender (S3), and whether age implies an appropriate value of experience (S4). We specify a mismatch between age and experience whenever experience exceeds age minus the minimum working age. Violation rates are reported in Table 4. TabSCM is able to outperform full diffusion models with lower violation rates, notably LLM-based methods, e.g., GReaT dominated this metric. The underlying transformer and attention-based architecture enhance contextualization through tokenization, enabling GReaT to generate highly realistic samples for these selected pairs of variables, closely followed by TabSCM.

| Method | (S1) | (S2) | (S3) | (S4) |
|---|---|---|---|---|
| Real | 2.24 | 0.00 | 0.07 | 7.60 |
| GReaT | $\mathbf{3.96 \pm 0.12}$ | $9.92 \pm 22.2$ | $\mathbf{0.01 \pm 0.04}$ | $7.82 \pm 0.04$ |
| CTGAN | $20.9 \pm 0.14$ | $55.1 \pm 0.31$ | $10.83 \pm 4.24$ | $\mathbf{4.79 \pm 0.02}$ |
| TabDDPM | $99.9 \pm 0.05$ | $0.38 \pm 0.01$ | $0.27 \pm 0.25$ | $28.9 \pm 0.01$ |
| TabSyn | $\underline{6.75 \pm 0.15}$ | $0.78 \pm 0.05$ | $0.28 \pm 0.25$ | $7.43 \pm 0.03$ |
| TabDiff | $8.62 \pm 0.30$ | $\underline{0.22 \pm 0.02}$ | $1.60 \pm 1.13$ | $7.69 \pm 0.05$ |
| GOGGLE | $49.7 \pm 0.24$ | $68.6 \pm 0.21$ | $48.9 \pm 42.59$ | OOM |
| TabSCM | $6.95 \pm 0.18$ | $\mathbf{0.00 \pm 0.00}$ | $\underline{0.11 \pm 0.10}$ | $\underline{7.36 \pm 0.05}$ |

Table 4: Reports violation rates (%) described in Section 6. **Bold** numbers indicate best performance; underlined values are second best results for each sanity check.

## 7 COUNTERFACTUAL INTERVENTIONS

**TabSCM allows for realistic counterfactual interventions and is able to generate out-of-distribution samples.** The Adult dataset contains demographic and employment-related information of individuals above the age of 17.

We generate out-of-distribution samples for the Adult dataset, and intervene for age $\mathrm{do}(x_{\mathrm{age}} = 16)$. We additionally intervene for relationship as this is a root node and set it to "own-child" to generate realistic samples. Measuring the quality of out-of-distribution samples is a non-standard task. Such OOD interventions are particularly valuable when stress-testing predictive models or exploring scenarios that are realistic but absent or underrepresented in the training data. We conducted qualitative assessments and checked whether marital status, education, and income are meaningful with respect to age. Figure 4 shows the marginal distribution of synthetic interventions obtained with TabSCM and real data for four variables. TabSCM is able to provide plausible out-of-distribution samples on individuals of age 16, the majority of samples have an education between 9-th and 12-th grade, have an income below 50k, and were never married.

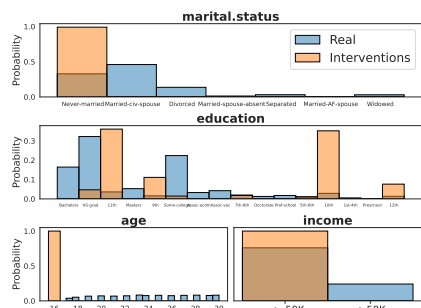

Figure 4: Marginal distribution of out-of-distribution counterfactual interventions of the Adult dataset with interventions $x_{\mathrm{age}} = 16$ and $x_{\mathrm{relationship}} =$"own-child". Real data is limited to $x_{\mathrm{age}} \geq 17$. We display the marginal distribution of martial.status, education, and income of real data. We only display $x_{\mathrm{age}} \leq 30$ in the histogram of the real age.

## 8 DISCUSSION AND CONCLUSION

We present TABSCM, a principled and efficient framework for generating realistic and causally coherent tabular data. In contrast to existing generative models that prioritize marginal similarity or downstream utility, TabSCM grounds the generation process in a structural causal model (SCM) instantiated from a completed partially directed acyclic graph. By orienting this graph into a valid DAG and fitting per-variable structural assignments, using conditional diffusion models for continuous variables and gradient-boosted trees for categorical ones, TabSCM factorizes the data-generating process into modular, interpretable components. This design ensures that samples are generated in topological order, preserving structural dependencies and avoiding rule violations by construction. Each conditional is a standalone, auditable model, which not only increases transparency but also supports fairness-aware modifications or policy-specific constraints. Importantly, while causal discovery from observational data is inherently imperfect, TabSCM is robust: through the oriented DAG, the method is designed to retain all parents, and the expressive conditional models compensate for minor misspecifications or spurious edges.

Empirical results across seven real-world datasets demonstrate that TabSCM consistently matches or outperforms state-of-the-art GAN, diffusion, and LLM-based generators in statistical fidelity, downstream predictive performance, and privacy. Moreover, it does so with significantly lower training time, up to $583\times$ faster than diffusion-only baselines, and with native support for semantically valid and counterfactually consistent samples. TabSCM further distinguishes itself as the only model in this space with built-in support for unit-level counterfactual inference, enabling "what-if" reasoning under hypothetical interventions.

Taken together, these results show that TabSCM achieves a rare combination of realism, interpretability through causal soundness, and efficiency. It repositions SCMs from theoretical constructs to practical tools for responsible data generation, particularly in high-stakes domains such as healthcare, finance, and policy modeling. By integrating structure with flexibility and causal reasoning with generative modeling, TabSCM offers a compelling foundation for future work on fair, explainable, and intervention-aware synthetic data.

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

## A  APPENDIX

### A.1  OUTLINE

To provide a comprehensive understanding of our experimental framework and results, the following presents additional experiments, detailed analysis, and the complete experimental setup. The structure is as follows:

- **Causal Discovery**: Outlines a brief description of each Causal discovery method used.
- **Ablation Study**: We investigate how the causal discovery method influences the quality of samples generated by TabSCM, and investigate the influence of training and diffusion steps on sample quality.
- **Additional Results**: We provide an additional qualitative assessment of the violation rates for the Housing dataset. We report $\alpha$-precision, $\beta$-recall, and detection score for each method and dataset, respectively.
- **Experimental setup**: We provide a detailed description of the experimental setup, including datasets used, data preprocessing applied, chosen hyperparameter settings for numerical and categorical nodes, and the full set of hyperparameters for each dataset.

### A.2  CAUSAL DISCOVERY

The starting point of TabSCM is a Completed Partially Directed Acyclic Graph (CPDAG) modeling the describing the inter-variable relationship of the tabular data. Unveiling unknown or hidden relationships between variables is a well-known and long-studied problem in Causal Discovery. Generally, Causal discovery describes the task of inferring the underlying causal structure among a set of variables. Unlike correlation-based methods, causal discovery aims to reveal the directionality of relationships, enabling robust predictions under interventions and distributional shifts. Several algorithms have been proposed to perform causal discovery under different assumptions. TabSCM utilizes three popular approaches, PC Spirtes et al. (2000), GES Chickering (2002), and NOTEARS Zheng et al. (2018).

- **PC**: Is a constraint-based method, testing conditional independencies among variables, it is lightweight and scalable, but sensitive to the choice of conditional independence test (CI-test) and threshold $\alpha$ applies
  **Parameter**:

- CI-test: "fisherz", "chisq"
- Significance level: $\alpha = 0.05$

- **GES**: Is a score-based method iteratively adding, removing, and reversing edges to maximize a scoring function

  **Parameter**:

  - Scoring function: "local_core_BIC"

- **NOTEARS**: Frames causal discovery as a continuous optimization problem, directly learning a weighted adjacency matrix representing a DAG.

  **Parameter**:

  - Loss funtion: $l_2$,
  - minimal weight: $w_{\min} = 0.1$,
  - $l_1$ regularization: $\lambda_1 = 0.05$

We use the implementation of PC and GES provided in *causal-learn*[4], and use the provided implementation of the linear version of NOTEARS[5].

PC is an efficient simple method to derive causal graphs, GES and NOTEARS can cope with complex tabular data and relationships.

### A.3 ABLATION STUDY

In the following, we conduct an ablation study of the causal discovery algorithm and the hyperparameters of TabSCM. We use the magic dataset, which is numerical heavy. We investigate how the causal discovery algorithm influences the statistical fidelity and downstream utility, and show how diffusion steps and training epochs affect training time and the data quality of generated samples. We will PC algorithm with CI_test={fisherz,chisq} with significance level $\alpha = \{0.01, 0.05, 0.1\}$. Additionally, we use NOTEARS with minimal weight $w_{\min} \in \{0.01, 0.2, 0.3\}$ and $\lambda_1 \in \{0.01, 0.05, 0.1\}$ The default epochs and diffusion steps are set to 200 and 2000.

#### A.3.1 INFLUENCE OF CAUSAL DISCOVERY ALGORITHM

We report the average error rate of column-wise density estimation in Table 5 and the error rate of the correlation estimation in Table 6. In this setting, a significance level $\alpha = 0.05$ reports the best results for both conditional independence tests. All parameter combinations lead to a similar density estimation and correlation error. Therefore, we observe that all parameter combinations result in similar downstream utility and DCR values, see Table 7.

We also report the error rates of column-wise density estimation and correlation estimation using NOTEARS for different values of $w_{\min}, \lambda_1$, see Tables 8 and 9. Generally, a stronger l1-regularization enforces sparsity of the estimated adjacency matrix which decreases the number of edges of the inferred causal graph, this can may oversimplify the interconnection of variables and increase the final correlation error between real and synthetic data, see Table 9. The minimal weight specifies the cutoff threshold of each edge, if an edge has a weight below this minimal threshold it is deleted, this also induces sparsity of the adjacency matrix. If this threshold is set to high, important edges may be deleted and are not represented in the synthetic data, thus increasing the correlation error eventually. In this experiment, a dense adjacency matrix ($w_{\min} = 0.01, \lambda_1 = 0.01$) reported minimal density estimation error and a significant decrease of the correlation error. Thus, this parameter combination also led to the best downstream utility.

#### A.3.2 INFLUENCE OF TRAINING AND DIFFUSION STEPS

In the following, we investigate how the number of epochs $n$ and the number of diffusion steps $t$ influence the statistical fidelity and downstream utility. We set $n = \{100, 250, 500, 1000\}$ and $t = \{t = 500, 1000, 1500, 2000\}$ for each run. We used the same DAG, which we constructed using NOTEARS with $\lambda_1 = 0.01, w_{\min} = 0.01$. We fitted TabSCM to the same training data with a

---

[4]https://github.com/py-why/causal-learn
[5]https://github.com/xunzheng/notears/tree/master

| CI_test | $\alpha = 0.01$ | $\alpha = 0.05$ | $\alpha = 0.1$ |
|---|---|---|---|
| fisherz | $4.44 \pm .09$ | $4.12 \pm .12$ | $4.36 \pm .09$ |
| chiqs | $4.37 \pm .09$ | $4.11 \pm .19$ | $4.26 \pm .06$ |

Table 5: The error rate (%) of column-wise density estimation (Eq.3). Lower values indicate more accurate estimation (superior results).

| CI_test | $\alpha = 0.01$ | $\alpha = 0.05$ | $\alpha = 0.1$ |
|---|---|---|---|
| fisherz | $7.21 \pm .40$ | $7.07 \pm .47$ | $7.08 \pm .80$ |
| chiqs | $7.28 \pm .56$ | $6.84 \pm .31$ | $7.26 \pm .58$ |

Table 6: The error rate (%) (Eq.4) of correlation estimation between column distribution of real and synthetic data for different parameter combinations.

| Parameter | | fisherz | | | chiqs | | |
|---|---|---|---|---|---|---|---|
| | **Real** | $\alpha = 0.01$ | $\alpha = 0.05$ | $\alpha = 0.1$ | $\alpha = 0.01$ | $\alpha = 0.05$ | $\alpha = 0.1$ |
| AUC ($\uparrow$) | 0.948 | $0.860 \pm .005$ | **0.866 ± .004** | $0.862 \pm .002$ | $\underline{0.856 \pm .007}$ | $0.865 \pm .007$ | $0.859 \pm .005$ |
| DCR | - | $0.312 \pm .007$ | $0.307 \pm .002$ | $0.315 \pm .001$ | $0.309 \pm .001$ | $0.305 \pm .001$ | $0.307 \pm .002$ |

Table 7: Results of the accuracy of classifiers/regressors trained on synthetic data evaluated on real test data. **Bold** numbers indicate best performance; underlined values are second best results for each dataset. DCR values are not highlighted.

| NOTEARS | $w_{\min} = 0.01$ | $w_{\min} = 0.2$ | $w_{\min} = 0.3$ |
|---|---|---|---|
| $\lambda_1 = 0.01$ | $4.20 \pm .12$ | $4.65 \pm .16$ | $4.76 \pm .06$ |
| $\lambda_1 = 0.05$ | $5.23 \pm .06$ | $3.87 \pm .08$ | $3.70 \pm .05$ |
| $\lambda_1 = 0.1$ | $5.94 \pm .14$ | $4.82 \pm .11$ | $6.03 \pm .18$ |

Table 8: The error rate (%) of column-wise density estimation (Eq. 3)). Lower values indicate more accurate estimation (superior results).

| NOTEARS | $w_{\min} = 0.01$ | $w_{\min} = 0.2$ | $w_{\min} = 0.3$ |
|---|---|---|---|
| $\lambda_1 = 0.01$ | $2.96 \pm .62$ | $8.26 \pm .15$ | $8.96 \pm .58$ |
| $\lambda_1 = 0.05$ | $9.64 \pm 1.12$ | $8.15 \pm .49$ | $8.25 \pm .98$ |
| $\lambda_1 = 0.1$ | $10.84 \pm .55$ | $9.89 \pm .57$ | $11.24 \pm .63$ |

Table 9: The error rate (%) (Eq. 4) of correlation estimation between column distribution of real and synthetic data for different parameter combinations.

| Parameter | | $w_{\min} = 0.01$ | | | $w_{\min} = 0.2$ | | | $w_{\min} = 0.3$ | | |
|---|---|---|---|---|---|---|---|---|---|---|
| | **Real** | $\lambda_1 = 0.01$ | $\lambda_1 = 0.05$ | $\lambda_1 = 0.1$ | $\lambda_1 = 0.01$ | $\lambda_1 = 0.05$ | $\lambda_1 = 0.1$ | $\lambda_1 = 0.01$ | $\lambda_1 = 0.05$ | $\lambda_1 = 0.1$ |
| AUC ($\uparrow$) | **0.948** | **0.925 ± .002** | $0.902 \pm .008$ | $0.906 \pm .003$ | $0.903 \pm .003$ | $0.903 \pm .004$ | $0.898 \pm .004$ | $0.897 \pm .005$ | $0.906 \pm .005$ | $0.903 \pm .006$ |
| DCR | - | $0.227 \pm .001$ | $0.370 \pm .001$ | $0.436 \pm .001$ | $0.391 \pm .002$ | $0.391 \pm .001$ | $0.457 \pm .002$ | $0.395 \pm .001$ | $0.389 \pm .001$ | $0.485 \pm .003$ |

Table 10: Accuracy of classifiers/regressors trained on synthetic data evaluated on real test data. **Bold** numbers indicate best performance; underlined values are second best results for each dataset. DCR values are not highlighted.

unique combination of epochs and diffusion steps. After fitting TabSCM, we sampled five synthetic datasets independently. We report the mean error of the density and correlation estimation with standard deviation in Tables 11 and 12. Increasing the number of epochs leads to a better density estimation error for all numbers of diffusion steps, we see the opposite behavior for the correlation

estimation. Increasing the number of diffusion steps leads to a better correlation estimation for the same number of epochs.

For all parameter combinations, the AUCs are above $0.920$, such that all samples show a high utility for augmenting the training set. In contrast, the AUC score of real data is $0.948$. Increasing the number of epochs improves the AUC score in general, while the number of diffusion steps does not primarily influence AUC results for the same number of epochs.

| $n/t$ | $t = 500$ | $t = 1000$ | $t = 1500$ | $t = 2000$ |
|---|---|---|---|---|
| $n = 100$ | $4.09 \pm 0.07$ | $4.09 \pm 0.11$ | $4.18 \pm 0.08$ | $4.31 \pm 0.08$ |
| $n = 250$ | $3.37 \pm 0.13$ | $3.65 \pm 0.11$ | $3.87 \pm 0.69$ | $3.92 \pm 0.09$ |
| $n = 500$ | $2.98 \pm 0.15$ | $2.86 \pm 0.14$ | $2.87 \pm 0.07$ | $3.09 \pm 0.11$ |
| $n = 1000$ | $2.71 \pm 0.15$ | $2.65 \pm 0.05$ | $2.51 \pm 0.08$ | $2.74 \pm 0.09$ |

Table 11: The error rate (%) of column-wise density estimation (Eq.3). Lower values indicate more accurate estimation.

| $n/t$ | $t = 500$ | $t = 1000$ | $t = 1500$ | $t = 2000$ |
|---|---|---|---|---|
| $n = 100$ | $3.06 \pm 0.36$ | $2.82 \pm 0.37$ | $2.95 \pm 0.09$ | $3.03 \pm 0.19$ |
| $n = 250$ | $3.17 \pm 0.48$ | $2.92 \pm 0.46$ | $3.07 \pm 0.69$ | $2.79 \pm 0.53$ |
| $n = 500$ | $3.29 \pm 0.33$ | $2.82 \pm 0.61$ | $2.81 \pm 0.08$ | $2.79 \pm 0.49$ |
| $n = 1000$ | $4.10 \pm 0.46$ | $3.52 \pm 0.30$ | $3.15 \pm 0.51$ | $3.29 \pm 0.29$ |

Table 12: The error rate (%) (Eq.4) of correlation estimation between column distribution of real and synthetic data for different parameter combinations.

| $n/t$ | $t = 500$ | $t = 1000$ | $t = 1500$ | $t = 2000$ |
|---|---|---|---|---|
| $n = 100$ | $0.923 \pm .003$ | $0.924 \pm .004$ | $0.920 \pm .002$ | $0.922 \pm .004$ |
| $n = 250$ | $0.931 \pm .002$ | $0.929 \pm .002$ | $0.930 \pm .002$ | $0.928 \pm .006$ |
| $n = 500$ | $0.930 \pm .003$ | $0.931 \pm .003$ | $0.931 \pm .002$ | $0.931 \pm .004$ |
| $n = 1000$ | $0.932 \pm .002$ | $0.934 \pm .003$ | $0.931 \pm .002$ | $0.932 \pm .004$ |

Table 13: Accuracy of classifiers/regressors trained on synthetic data evaluated on real test data.

In the next experiment, we vary the number of epochs $n = \{100, 200, 300, 400, 500, 600, 700, 800, 900, 1000\}$ while we fix the number of diffusion steps $t = 500$. Increasing the number of epochs decreases the error of the density estimation, but this comes at a cost. This cost is runtime, which is increasing for larger epochs. We illustrate the results in Figure 5.

## A.4 ADDITIONAL METRICS

### A.4.1 HOUSING

We conduct a sanity check to determine if the longitude and latitude pair indicate a location within the state of California. For the specific boundary[6] data, the real training data had an inherent violation rate of $2.24\%$. This can happen given that we collected the boundary data independently. Figure 6 shows the qualitative assessment of boundary violations for each method. Additionally, we report the 2-Wasserstein distance between the joint distribution of longitude, latitude of real and synthetic data. The Wasserstein distance is an optimal transport-based metric that measures the cost of transforming one source distribution into a certain target distribution. It goes beyond a single point measure such as the KS-distance. TabSCM shows the lowest 2-Wasserstein distance, closely followed by TabSyn.

---

[6]https://github.com/PublicaMundi/MappingAPI/blob/master/data/geojson/us-states.json

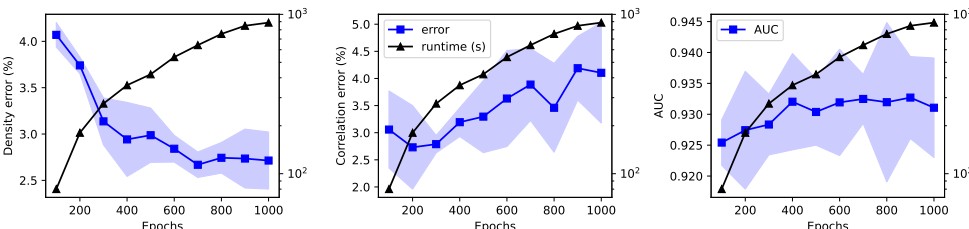

Figure 5: Illustrates how the number of epochs influences density estimation error (left), correlation error (middle), and the AUC score (right) with respect to the runtime, which is measured in seconds.

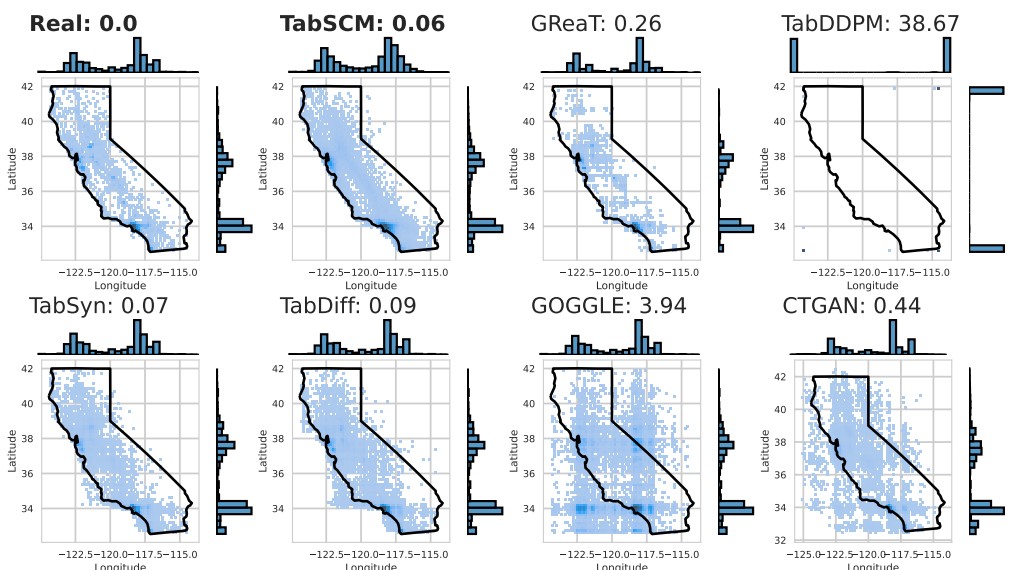

Figure 6: Comparison of original and generated samples from the California Housing dataset. Joint histogram plots illustrate the relationship between the highly correlated variables, Latitude and Longitude. The number beside the title of each subplot indicates the 2-Wasserstein distance between synthetic and real data samples. The black outline indicates the true boundary of the state of California

### A.4.2 $\alpha$-PRECISION AND $\beta$-RECALL

We follow prior work Liu et al. (2023); Zhang et al. (2024); Shi et al. (2025) also evaluating data quality on a higher-order statistics assessing the models capability to capture the joint distribution. As we have previously outlined, solely evaluating MLE without context might undermine privacy, but it also ignores less informative columns. Therefore, we follow Zhang et al. (2024) and evaluate the adopted $\alpha$-precision and $\beta$-recall Alaa et al. (2022):

- $\alpha$-precision: how realistic (faithful) synthetic samples are

- $\beta$-recall how well synthetic data covers the real data distribution.

In general, diffusion only models including TabSyn and TabDiff show the best $\alpha$-precision and $\beta$-recall, TabSCM is outperforming both SoTA methods in small scale Diabetes and large scale Loan dataset and delivers competitive result for all other datasets. The results are reported in Tables 14 and 15.

---

[6]Boundary obtained from `https://github.com/PublicaMundi/MappingAPI/blob/master/data/geojson/us-states.json`

| Method | Adult | Beijing | Diabetes | HELOC | Housing | Loan | Magic |
|---|---|---|---|---|---|---|---|
| GReaT | $0.558 \pm .001$ | $0.974 \pm .003$ | - | $0.891 \pm .004$ | $0.899 \pm .004$ | - | $0.839 \pm .006$ |
| CTGAN | $0.780 \pm .003$ | $0.929 \pm .002$ | $0.885 \pm .014$ | $0.957 \pm .002$ | $0.950 \pm .003$ | $0.869 \pm .002$ | $0.768 \pm .003$ |
| TabDDPM | $0.958 \pm .001$ | $\mathbf{0.987 \pm .001}$ | $0.866 \pm .027$ | $0.919 \pm .004$ | - | $0.450 \pm .002$ | $0.981 \pm .006$ |
| TabSyn | $\mathbf{0.991 \pm .002}$ | $0.979 \pm .002$ | $\mathbf{0.978 \pm .005}$ | $\underline{0.978 \pm .003}$ | $\mathbf{0.994 \pm .001}$ | $\underline{0.946 \pm .002}$ | $\underline{0.992 \pm .004}$ |
| TabDiff | $0.981 \pm .002$ | $\underline{0.985 \pm .001}$ | $\underline{0.972 \pm .006}$ | $0.975 \pm .002$ | $0.990 \pm .002$ | $\mathbf{0.994 \pm .004}$ | $\mathbf{0.994 \pm .004}$ |
| GOGGLE | $0.541 \pm .002$ | $0.944 \pm .001$ | $0.372 \pm .008$ | $0.908 \pm .007$ | $0.947 \pm .002$ | OOM | $0.943 \pm .002$ |
| TabSCM (Ours) | $\underline{0.982 \pm .001}$ | $0.981 \pm .002$ | $0.949 \pm .013$ | $\mathbf{0.984 \pm .003}$ | $\underline{0.991 \pm .002}$ | $0.938 \pm .002$ | $0.955 \pm .003$ |

Table 14: Comparison of $\alpha$-precision scores. **Bold** values represents the best score, and underlined values are second best for each dataset. Higher values indicate superior results.

| Method | Adult | Beijing | Diabetes | HELOC | Housing | Loan | Magic |
|---|---|---|---|---|---|---|---|
| GReaT | $0.487 \pm .002$ | $0.432 \pm .003$ | - | $0.455 \pm .004$ | $\underline{0.425 \pm .002}$ | - | $0.343 \pm .002$ |
| CTGAN | $0.274 \pm .001$ | $0.394 \pm .001$ | $0.087 \pm .012$ | $0.111 \pm .002$ | $0.260 \pm .003$ | $0.055 \pm .002$ | $0.09 \pm .003$ |
| TabDDPM | $\underline{0.488 \pm .001}$ | $0.555 \pm .001$ | $0.059 \pm .008$ | $\mathbf{0.541 \pm .006}$ | - | $0.031 \pm .001$ | $0.471 \pm .006$ |
| TabSyn | $0.483 \pm .002$ | $\underline{0.568 \pm .002}$ | $0.226 \pm .005$ | $\underline{0.490 \pm .003}$ | $\mathbf{0.429 \pm .002}$ | $\underline{0.103 \pm .001}$ | $\mathbf{0.478 \pm .002}$ |
| TabDiff | $\mathbf{0.545 \pm .004}$ | $\mathbf{0.582 \pm .001}$ | $\underline{0.232 \pm .009}$ | $0.439 \pm .002$ | $0.422 \pm .003$ | $0.072 \pm .003$ | $\underline{0.474 \pm .003}$ |
| GOGGLE | $0.07 \pm .001$ | $0.05 \pm .001$ | $0.027 \pm .004$ | $0.293 \pm .003$ | $0.072 \pm .002$ | OOM | $0.204 \pm .002$ |
| TabSCM (Ours) | $0.289 \pm .001$ | $0.503 \pm .001$ | $\mathbf{0.254 \pm .020}$ | $0.450 \pm .003$ | $0.397 \pm .001$ | $\mathbf{0.126 \pm .002}$ | $0.312 \pm .002$ |

Table 15: Comparison of $\beta$-recall scores. **Bolde** values represents the best scoreand, and underlined values are second best for each dataset. Higher values indicate superior results.

### A.4.3 DETECTION SCORE

We apply a Classifier Two Sample Test (C2ST) to quantify how difficult it is to distinguish real from synthetic data. We follow the setup given by sdmetrics[7], where a label gets assigned for each row of real and synthetic tabular data, both datasets are randomly split into training and validation set, a classifier is trained and evaluated on the validation set, then this procedure is repeated for different training and validation splits. The final score is based on the average $\overline{\text{AUC}}$ of the ROC across the different splits,

$$\text{C2ST} = 1 - (2 \cdot \overline{\text{AUC}} - 1).$$

This score is maximized $\text{C2ST} = 1$ when real and synthetic data are indistinguishable to the classifier, which corresponds to random guessing whether a sample is real or synthetic and vice versa. We calculate the detection score for five different synthetic datasets sampled by each method for each dataset, respectively. The average score and standard deviation are reported in Table 16.

### A.5 EXPERIMENTAL SETUP

The following covers the datasets used, data preprocessing applied, and the hyperparameter specification of TabSCM.

---

[7] https://docs.sdv.dev/sdmetrics/metrics/metrics-in-beta/detection-single-table

| Method | Adult | Beijing | Diabetes | HELOC | Housing | Loan | Magic |
|---|---|---|---|---|---|---|---|
| GReaT | $0.532 \pm .004$ | $0.621 \pm .004$ | - | $0.517 \pm .003$ | $0.776 \pm .002$ | - | $.0419 \pm .004$ |
| CTGAN | $0.622 \pm .002$ | $0.827 \pm .003$ | $0.481 \pm .036$ | $0.729 \pm .005$ | $0.809 \pm .003$ | $0.451 \pm .001$ | $0.634 \pm .003$ |
| TabDDPM | $0.955 \pm .005$ | $0.953 \pm .003$ | $0.949 \pm .029$ | $0.904 \pm .009$ | $0.04 \pm .017$ | $0.273 \pm .002$ | $0.989 \pm .006$ |
| TabSyn | $\underline{0.979 \pm .006}$ | $0.944 \pm .001$ | $0.985 \pm .008$ | $\mathbf{0.934 \pm .007}$ | $\underline{0.992 \pm .005}$ | $0.709 \pm .001$ | $0.994 \pm .003$ |
| TabDiff | $\mathbf{0.985 \pm .002}$ | $\mathbf{0.959 \pm .005}$ | $\underline{0.999 \pm .002}$ | $\underline{0.924 \pm .007}$ | $0.968 \pm .006$ | $\mathbf{0.925 \pm .002}$ | $\mathbf{0.998 \pm .001}$ |
| GOGGLE | $0.112 \pm .004$ | $0.352 \pm .002$ | $0.001 \pm .001$ | $0.790 \pm .008$ | $0.712 \pm .001$ | OOM | $0.855 \pm .005$ |
| TabSCM (Ours) | $0.869 \pm .002$ | $\underline{0.953 \pm .003}$ | $\mathbf{0.999 \pm .0001}$ | $0.909 \pm .006$ | $\mathbf{0.995 \pm .004}$ | $\underline{0.902 \pm .001}$ | $0.889 \pm .004$ |

Table 16: Reports C2ST score. Lower values indicate a more accurate estimation (superior results). **Bold** numbers indicate best performance; underlined values are second best results for each dataset.

### A.5.1 DATASETS

We use a total of seven real-world datasets covering classification and regression tasks from various application domains. The data covers large-scale datasets (>250k samples) and small-scale datasets (<1000 samples). Adult, Beijing, Diabetes, and Magic are from the UCI Machine Learning Repository[8], Housing[9] from sklearn datasets, and Loan[10] and HELOC[11] from kaggle. Below, we provide a comprehensive overview of the datasets used. We denote the number of numerical columns #Num, the number of categorical columns #Cat. #Max Cat stands for the number of categories of the categorical column with the most categories.

| Dataset | # Rows | # Num | # Cat | # Max Cat | # Train | # Val | #Test | Taks |
|---|---|---|---|---|---|---|---|---|
| Adult | $48,842$ | 6 | 9 | 42 | $28,943$ | $3,618$ | $16,281$ | Classification |
| Beijing | $43,824$ | 7 | 5 | 31 | $35,058$ | $4,383$ | $4,383$ | Regression |
| Diabetes | 520 | 1 | 16 | 2 | 416 | 52 | 52 | Classification |
| HELOC | $10,459$ | 23 | 1 | 2 | $8,367$ | $1,046$ | $1,046$ | Classification |
| Housing | $20,640$ | 8 | 1 | 52 | $16,512$ | $2,046$ | $2,046$ | Regression |
| Loan | $252,000$ | 2 | 10 | 317 | $201,600$ | $25,200$ | $25,200$ | Classification |
| Magic | $19,019$ | 10 | 1 | 2 | $15,215$ | $1,902$ | $1,902$ | Classification |

Table 17: Overview of the datasets.

### A.5.2 DATA PREPROCESSING

Following prior work Kotelnikov et al. (2023); Zhang et al. (2024), we fill missing numerical data with the average column value of the corresponding column, and introduce an additional category for missing values of categorical columns. The baseline methods transform numerical columns using a QuantileTransformer[12], and deploy OneHotEncoding for categorical columns[13]. TabSCM transforms numerical data with the StandardScaler[14], and converts categorical columns with the LabelEncoder[15]. For columns with many categories, using Label encoding over OneHot encoding reduces the overall data matrix.

### A.5.3 HYPERPARAMETER SETTING

Below, we report the hyperparameter settings for the surrogate regression models for numerical nodes and the classification models for categorical nodes. Additionally, we report the hyperparameter set including the causal discovery method in Table 18.

**Numerical nodes** For numerical variables, we deploy a conditional diffusion model, implemented with an MLP-based U-Net with,

- *Hidden Dim*: 256,

- *Number of epochs*: epochs $\in \{200, 500, 1000\}$,

---

[8]https://archive.ics.uci.edu/datasets
[9]https://scikit-learn.org/stable/modules/generated/sklearn.datasets.fetch_california_housing.html
[10]https://www.kaggle.com/datasets/subhamjain/loanprediction-based-on-customer-behavior/data
[11]https://www.kaggle.com/datasets/averkiyoliabev/homeequity-line-of-creditheloc
[12]https://scikit-learn.org/stable/modules/generated/sklearn.preprocessing.QuantileTransformer.html
[13]https://scikit-learn.org/stable/modules/generated/sklearn.preprocessing.OneHotEncoder.html
[14]https://scikit-learn.org/stable/modules/generated/sklearn.preprocessing.StandardScaler.html
[15]https://scikit-learn.org/stable/modules/generated/sklearn.preprocessing.LabelEncoder.html

- *Diffusion steps*: $t \in \{500, 1000, 1500\}$,

- *Batch Size*: 512,

- *Learning Rate*: 0.001,

- *Optimizer*: Adam

**Categorical nodes**   Our experiment showed that tree-based classification models with gradient boosting matched the performance of categorical diffusion models, thus we deploy an XGBoost classifier Chen & Guestrin (2016) with:

- *Tree method*: "hist",

- *Learning Rate*: 0.2,

- *Max Depth*: 30,

- *Num. Boosting Rounds*: 500,

- *L1-reg*: 1.5,

- *L2-reg*: 1.5

Running a grid search for the hyperparameter may lead to superior results as reported.

| Dataset | Epochs | Diffusion steps | Causal Discovery | Weight Threshold | CI-Test | $\alpha$ |
|---------|--------|-----------------|------------------|------------------|---------|----------|
| Adult | 500 | 500 | GES | – | – | – |
| Beijing | 500 | 500 | NOTEARS | 0.1 | – | – |
| Diabetes | 500 | 500 | GES | – | – | – |
| HELOC | 1000 | 500 | NOTEARS | 0.1 | – | – |
| Housing | 500 | 1000 | GES | – | – | – |
| Loan | 500 | 1500 | PC | – | fisherz | 0.05 |
| Magic | 200 | 2000 | NOTEARS | 0.01 | – | – |

Table 18: Hyperparameter settings of TabSCM for each dataset.

| Method | Adult | Beijing | Diabetes | Heloc | Housing | Loan | Magic |
|--------|-------|---------|----------|-------|---------|------|-------|
| GReaT | 71.52 | 72.70 | 168.68 | 96.85 | 30.80 | 168.93 | 24.05 |
| TabDDPM | 33.25 | 58.19 | 35.37 | 28.85 | 35.75 | 32.02 | 30.25 |
| TabSyn | 36.51 | 32.35 | 15.99 | 31.55 | 25.79 | 127.16 | 38.72 |
| TabDiff | 394.27 | 338.41 | 115.68 | 85.61 | 116.22 | 1999.51 | 118.71 |
| GOGGLE | 309.09 | 492.42 | 1.31 | 26.41 | 62.89 | OOM | 17.63 |
| TabSCM (Ours) | **7.08** | **11.46** | **0.20** | **24.15** | **5.01** | **17.45** | **2.44** |

Table 19: Training time (in minutes) for each method across datasets. **Bold** values indicate best performance, and underlined values are second best for each dataset.

