# OpenReview forum: "Connecting the Dots: Generating Realistic Tabular Data with Structural Causal Models"
_ICLR.cc/2026/Conference — Submitted to ICLR 2026_

### Official Review · Reviewer_BjkV · 2025-10-27

**Soundness:** 2
**Presentation:** 1
**Contribution:** 2
**Rating:** 2
**Confidence:** 4

**Summary:**

This paper introduces **TabSCM**, a generative framework for **realistic tabular data synthesis** grounded in *Structural Causal Models (SCMs)*. Unlike existing diffusion- or GAN-based tabular generators that model joint distributions without causal semantics, TabSCM explicitly represents causal dependencies among variables through an estimated or provided DAG. The method first constructs or orients a causal graph, fits marginal distributions for exogenous variables, and then learns conditional mechanisms for each node using flexible estimators (e.g., neural networks, gradient-boosted trees, or diffusion models) following a topological order. Sampling proceeds ancestrally, enabling *interventional* and *counterfactual* generation in addition to standard synthetic data.

Empirical evaluations on multiple mixed-type tabular datasets demonstrate that TabSCM outperforms non-causal baselines such as CTGAN, TabDDPM, TabDiff, and GReaT in both statistical fidelity (distributional similarity) and structural preservation metrics. The paper argues that embedding causal structure enhances both realism and controllability of synthetic data. Overall, the contribution lies in bridging **causal modeling** and **tabular data generation**, showing that SCM-based synthesis can yield higher-quality and semantically consistent data samples.

**Strengths:**

* The paper frames tabular data generation explicitly through a structural causal model: learn a DAG, fit one mechanism per node conditioned on its parents, and sample ancestrally. This gives a principled way to generate data that is intervention-aware and can (at least in principle) support counterfactual-style queries, which most baselines cannot do.

* The approach is modular and auditable. Different conditional models are used for different variable types (e.g., diffusion-style models for continuous variables, boosted trees for categorical ones), which makes individual causal relationships inspectable and replaceable — useful for enforcing constraints or fairness.

* The evaluation goes beyond marginal fidelity and downstream accuracy, and looks at semantic validity (e.g., checking for impossible combinations like “wife” with male gender). This targets a real pain point in synthetic tabular data: generating rows that are plausible, not just statistically close.

**Weaknesses:**

1. **Method clarity and technical correctness**

   The paper says it takes a CPDAG from causal discovery, orients it into a DAG, removes cycles, and then uses that DAG to define the parent sets $PA_i$ for each node. It never explains how undirected edges are oriented, how ambiguities in the Markov equivalence class are resolved, or how cycles are detected and broken. Those steps define $PA_i$, so they are part of the method, not an implementation detail.

   The paper also claims that in its “topological order,” $f_{\pi^{-1}(1)} \prec f_{\pi^{-1}(2)}$ means “$\pi^{-1}(1)$ is a parent of $\pi^{-1}(2)$.” A topological order does not guarantee direct parenthood; it only guarantees that no node appears before its descendants. Treating “earlier in order” as “is a parent of” is incorrect and later blurs parents vs. arbitrary ancestors.

   The section called “Counterfactual reasoning” opens by describing $do(X{=}x)$ interventions (population-level) but presents them as counterfactuals. Counterfactuals are unit-level queries, not just $do(\cdot)$ interventions. The section also mixes basic SCM ingredients (structural assignments and exogenous noise) into the middle of the method instead of stating them clearly in the setup.

   Finally, the paper defines a numerical correlation error $e_{\text{corr}}^{\text{num}}$ but drops the $\tfrac{1}{2}$ normalization factor used in Zhang et al. 2024 [1], Eq. (37), which keeps the average absolute Pearson correlation gap in $[0,1]$ and makes it directly comparable across datasets and with categorical correlation errors. Changing that scaling without comment makes the reported metric harder to compare and can affect which model appears best.

2. **Counterfactual claims are not operationally supported**

   The paper claims support for “unit-level counterfactuals,” but does not specify an abduction step to infer per-sample exogenous noise $\hat{E} \sim P(E \mid X = x^{\text{obs}})$ for a given individual. In standard structural causal model semantics, answering a counterfactual for one specific record requires:

   1. abduction, to infer that individual’s latent noise $\hat{E}$,
   2. action, to apply $do(\cdot)$ on the mechanism of interest,
   3. prediction, to resimulate descendants using the same $\hat{E}$.

      Here, node mechanisms are implemented as conditional diffusion models (continuous children) or boosted trees (categorical children). The paper does not explain how to invert these mechanisms to recover $\hat{E}$ for a specific record, so abduction is not actually implemented.

      As a result, examples like “set age = 16 and regenerate the rest of the row” are interventional samples $p(\cdot \mid do(\text{age}{=}16))$, not per-person counterfactuals. Existing SCM toolkits such as DoWhy-GCM already model each node as a structural mechanism on a DAG and explicitly support abduction–action–prediction for individual-level counterfactuals Blöbaum et al. 2022 [2]. Work on diffusion models as causal mechanisms also treats diffusion generators as structural assignments that can answer interventions and counterfactuals once exogenous noise is identified Chao et al. 2024 [5]. The paper gestures at this idea but does not provide the actual inference step.

3. **Positioning / novelty**

   The pipeline in the paper is: estimate a causal graph; fit one conditional generator per node given $PA_i$; then sample ancestrally in topological order and answer $do(\cdot)$ queries by clamping variables. This is essentially the standard graphical causal model workflow: libraries like DoWhy-GCM learn one mechanism per node on a DAG and then generate interventional and (with abduction) counterfactual samples Blöbaum et al. 2022 [2].

   The main difference claimed here is the use of strong per-node generators (diffusion models for continuous variables, boosted trees for categorical ones) and the framing as a tabular data generator. But expressive diffusion-based tabular generators that aim to preserve column correlations and downstream task performance already exist, e.g. TabSyn Zhang et al. 2024 [1]. Recent work also treats diffusion models directly as causal mechanisms for interventions and counterfactuals Chao et al. 2024 [5].

   The paper does not clearly separate what is genuinely new (engineering integration choices) from what is already standard practice in SCM-based generation.

4. **Baselines and evaluation scope**

   The experiments compare against popular tabular generators (CTGAN, TabDDPM, TabSyn, GReaT) and report fidelity, downstream task performance, and privacy. However, they do not compare against causal/tabular baselines that already try to respect a graph or generate data under causal constraints, such as:

   * DoWhy-GCM, which fits a DAG with per-node mechanisms and supports interventional and counterfactual queries Blöbaum et al. 2022 [2]
   * VACA, which learns latent exogenous variables consistent with a DAG and is explicitly designed to support intervention and counterfactual queries via abduction–action–prediction Sánchez-Martín et al. 2022 [3]
   * Causal-TGAN, which conditions tabular generation on a learned causal graph Wen et al. 2022 [4]
     Without these, it’s hard to tell whether the reported “causal coherence” improvements are due to enforcing a graph at all, or mostly due to the strength of the chosen per-node learners.

     In addition, the paper presents timing numbers but does not include a scaling study for structure: there is no analysis of memory or compute as the number of columns or the maximum in-degree grows. Since the approach trains and stores one mechanism per node, scalability with wide, high-dependency tables is an open question.

---

**References**

[1] H. Zhang, J. Zhang, Z. Shen, B. Srinivasan, X. Qin, C. Faloutsos, H. Rangwala, G. Karypis. 2024. TabSyn: Mixed-Type Tabular Data Synthesis with Score-Based Diffusion in Latent Space. ICLR 2024.

[2] P. Blöbaum, P. Götz, K. Budhathoki, A. A. Mastakouri, D. Janzing. 2022. DoWhy-GCM: An Extension of DoWhy for Causal Inference in Graphical Causal Models.

[3] P. Sánchez-Martín, M. Rateike, I. Valera. 2022. VACA: Designing Variational Graph Autoencoders for Causal Queries. AAAI 2022.

[4] B. Wen, L. O. Colon, K. P. Subbalakshmi, R. Chandramouli. 2022. Causal-TGAN: Modeling Tabular Data Using Causally-Aware GANs. ICLR 2022 Workshop on Deep Generative Models for Highly Structured Data.

[5] P. Chao, P. Blöbaum, S. Patel, S. P. Kasiviswanathan. 2024. Diffusion Models as Causal Mechanisms for Interventions and Counterfactuals.

**Questions:**

1. **Graph construction / parent sets**

   You say you start from a CPDAG, “orient edges to a DAG,” and “remove cycles,” and then you use that DAG to define the parent sets $PA_i$ for each node.

   How exactly do you (a) orient undirected edges, (b) resolve ambiguities in the Markov equivalence class, and (c) break any directed cycles that appear?

   When breaking cycles, do you ever delete edges, and if so, how do you decide which edges to drop?

   (Since $PA_i$ defines every node’s conditional generator, this procedure seems like part of the method, but it is not described.)

2. **Counterfactual generation procedure**

   You claim TabSCM supports “unit-level counterfactuals.” For a given observed record $x$, what is the exact algorithm you run to produce its counterfactual under $do(A{=}a')$?

   In particular: how do you infer that specific record’s latent noise / exogenous state (the abduction step), and how do you ensure that same latent state is reused after changing $A$?

   Today the examples in the paper (e.g. setting `age = 16` and regenerating the row) look like draws from $p(\cdot \mid do(\text{age}{=}16))$ at the population level. Are you in fact producing population-level interventional samples, or true per-individual counterfactuals?

3. **Scalability / complexity**

   The method fits one conditional generator per node (diffusion for continuous children, boosted trees for categorical children) and samples through all of them in topological order.

   How does training time, memory footprint, and sampling time scale with the number of columns and with node in-degree / graph density?

   Do you have experiments where you sweep graph size / sparsity, or stress-test wide, high-dependency tables, to show that the approach remains tractable?

---

### Official Review · Reviewer_xwiG · 2025-10-28

**Soundness:** 2
**Presentation:** 2
**Contribution:** 2
**Rating:** 2
**Confidence:** 4

**Summary:**

The author propose TabSCM, a generative model that combines causal discovery with diffusion models and gradient boosted trees to generate realistic, causal consistent and privacy-preserving samples of mixed-type tabular data.
The generative model also enables counterfactual sampling and interventional distribution modeling by ancestral sampling. The model is validated in utility, density estimation, privacy and time consumption on several benchmarking tabular datasets.
Results showed that the model is competitive with other SOTA methods based in GANs, Diffusion models and LLMs in utility and density estimation, while it takes less training time and enable causal inference.

In general, the methodology is well explained and has several correct points from a practical point of view. However, some of the main claims of the paper cannot be guaranteed, due to the mispecificity of the extracted causal graph. In addition, the literature is poorly covered, since there have been several advances in the field of generative causal models in the last years. To be accepted, the paper should 1) cover the relevant literature and include several methods in the comparison, 2) extend the experiments to include causal validation of counterfactuals and interventional distribution, 3) discuss and cover the implications of misspecificity of the extracted causal graph. In its current state, I cannot recommend to accept the paper.

**Strengths:**

- Overall, the purpose of the paper is clear, and the proposed method is well explained, so both the problem and the solution can be understood easily. Both problem setup, method (except the part of orienting the edges), experimental setup are well explained.

- The contribution of workflow,  causal discovery + causal inference via Generative model, is fair and useful from a practical point of view. The inclusion of boosted trees for categorical variables makes the model more flexible.

- The experiments are well done in general, systematic, and they present a comparison of several useful metrics: utility, density matching, privacy and time consumption. The validation is well defined and covers some important questions about the quality of the synthetic data generated.

**Weaknesses:**

> Major problems

My main concern is related with the literature review. The authors completely ignored all the litarature about Causal Generative Models (CGM), which is, basically, what TabSCM is. I will list here a few [1-10]. This is a severe concern for several reasons:

- TabSCM is a causal generative model which basically trains one generative model per variable, conditioning in the parents in the causal graph. That approach is very common in causality, and there exist huge libraries that already implement this, e.g. PyWhy [11]. Therefore, the contributions of `lines 79 - 86` are overstated, since 1) the comparison of non-causal with causal generative models is unfair and CGMs should have been into account in validations, and 2) the insighs of counterfactual interventions are inherent properties of the CGMs.

- Specially, there exist one Diffusion based causal model that follows a very similar approach, DCM [1]. Although [1] assumes that the causal graph is known, and it does not handle categorical variables, the similarities are too important so the model has to be mentioned and compared.

- In terms of time efficiency, I would like to see a comparative with other generative models not based on GANs or Diffusion, as normalizing flows [12], that already generate the synthetic i-th covariate from the previous. Also with its extension to include causal DAGs, causal normalizing flows [2]. Note that, although those models are for continuous variables, authors argue that adding noise to discrete variables should work.

- The results are not very conclusive. In density estimation, TabSCM is the best model in 3 of the 7 dataset, while it achieves the best utility in 2 of the 7 datasets. Therefore, we could say that is competitive, but not a better generative model.

I also have other concerns that are important to address:

- One of the biggest claim is that TabSCM is 583$\times$ faster than other generative models. However, the whole framework implies causal a previous process of causal discovery. A fair comparison should include the time that causal discovery takes, since it is needed for TabSCM to work.

- In`line 15`, the authors state that TabSCM orients the edges of a Completed Partially DAG, but I cannot find that stel in the methodology. It seems to be arbitrary. Can the authors provide any insights about this process?

- The counterfactual reasoning, although is stated as one of the contributions of the paper, and there is an experiment in which the operation is shown, is not validated. That is, the authors argue that the model can perform counterfactual inference, but there is not theoretic guarantee of  its validity and the experiments do not compare the generated counterfactuals with any ground truth.

- Related with the previous point, the authors claim that counterfactuals can be estimated for discrete/categorical variables. However, a transformation from $f(\epsilon, X_{i<j})\rightarrow X_j$, when $X_j$ is discrete, cannot be bijective from $\epsilon$ to $X_j$. Therefore, how can counterfactuals be calculated in theory? Authors should address this limitation, at least with experiments that support in practice the idea that counterfactuals can be approximated.

- Both interventional distribution and counterfactuals rely, not only on the generative model, but also on the causal discovery method used to extract the causal graph. If the causal discovery method and the generative model use the same information, errors commited in the first stage will be propagated to the second stage. On the other hand, if TabSCM orients the edges, as postulated in `line 15`, how to ensure that edges are well oriented? The data may present some underterminancies about causal direction that are not recoverable with the data itself, and external causal knowledge is needed. If causal directions are misspecified, intervenional distributions and counterfactuals will be biased. Causal consistency, i.e., $\hat{G}=G$, where $G$ is the true causal graph, does not seem to be guaranteed.

- When measuring privacy preservation, distance to closest records can be a good sanity chek, but is insufficient to measure privacy leakage [13]. Other metrics or approaches such as differential privacy should be used to have guarantees.


> Summary

Although the proposal could be interesting from a practical point of view, the lack of a fair comparison with models of the same class (CGMs) prevents me from accepting the paper. The methodological contribution is valid: causal discovery + causal generative model. However, causal consistency is not guaranteed with this process, and therefore all the advantages of the model are only conjetural. Results shows competitive performance, but without causal guarantees, that competitive performance is not enough to accept the paper.

> References

[1] Chao, P., Blöbaum, P., & Kasiviswanathan, S. P. (2023). Interventional and counterfactual inference with diffusion models. arXiv preprint arXiv:2302.00860, 4, 16.

[2] Javaloy, A., Sánchez-Martín, P., & Valera, I. (2023). Causal normalizing flows: from theory to practice. Advances in Neural Information Processing Systems, 36, 58833-58864.

[3] Ilyes Khemakhem, Ricardo Pio Monti, Robert Leech, and Aapo Hyvarinen. Causal Autoregressive Flows. In Arindam Banerjee and Kenji Fukumizu, editors, The 24th International Conference on Artificial Intelligence and Statistics, AISTATS 2021
2021.

[4] Almodóvar, A., Javaloy, A., Parras, J., Zazo, S., & Valera, I. (2025). DeCaFlow: A Deconfounding Causal Generative Model. arXiv preprint arXiv:2503.15114.

[5]  Alvaro Parafita and Jordi Vitria. Estimand-agnostic causal query estimation with deep causal graphs. IEEE Access, 10:71370–71386, 2022

[6] Nick Pawlowski, Daniel Coelho de Castro, and Ben Glocker. Deep Structural Causal Models for
Tractable Counterfactual Inference. In Hugo Larochelle, Marc’Aurelio Ranzato, Raia Hadsell, Maria-Florina Balcan, and Hsuan-Tien Lin, editors, Advances in Neural Information Processing Systems 33: Annual Conference on Neural Information Processing Systems 2020, NeurIPS 2020, December 6-12, 2020, virtual, 2020.

[7] Md. Musfiqur Rahman and Murat Kocaoglu. Modular Learning of Deep Causal Generative Models for High-dimensional Causal Inference. In Forty-first International Conference on Machine Learning, ICML 2024, Vienna, Austria, July 21-27, 2024

[8] Patrik O. Hoyer, Dominik Janzing, Joris M. Mooij, Jonas Peters, and Bernhard Scholkopf. Nonlinear causal discovery with additive noise models. In Daphne Koller, Dale Schuurmans, Yoshua Bengio, and Leon Bottou, editors, Advances in Neural Information Processing Systems 21, Proceedings of the Twenty-Second Annual Conference on Neural Information Processing Systems, Vancouver, British Columbia, Canada, December 8-11, 2008

[9] Pablo Sanchez-Martin, Miriam Rateike, and Isabel Valera. VACA: Designing Variational Graph Autoencoders for Causal Queries. In Thirty Sixth AAAI Conference on Artificial Intelligence, AAAI 2022

[10] Matej Zecevic, Devendra Singh Dhami, Petar Velivckovi ´ c, and Kristian Kersting. Relating graph neural networks to structural causal models. ArXiv preprint, abs/2109.04173, 2021.

[11] Patrick Blöbaum, Peter Götz, Kailash Budhathoki, Atalanti A. Mastakouri, Dominik Janzing. DoWhy-GCM: An extension of DoWhy for causal inference in graphical causal models. 2022.

[12] Papamakarios, G., Nalisnick, E., Rezende, D. J., Mohamed, S., & Lakshminarayanan, B. (2021). Normalizing flows for probabilistic modeling and inference. Journal of Machine Learning Research, 22(57), 1-64.

[13] Kaabachi, B., Despraz, J., Meurers, T., Otte, K., Halilovic, M., Kulynych, B., ... & Raisaro, J. L. (2025). A scoping review of privacy and utility metrics in medical synthetic data. NPJ digital medicine, 8(1), 60.

**Questions:**

I have also some minor concerns that will improve the readability and the communication of the paper:

- The way to cite some papers, specially in the introduction, does not make easy the reading. The citations that are not part of the text should have parenthesis. Otherwise, the reader may get confused.

- The literature about synthetic data generation in the introduction (`line 32 to 42`) is biased towards GAN-based approaches. All those papers are related to GANs, and do not reflect the diversity of models that have arisen in the last 10 years. Is there any reason to not cover all those contributions?

- Do the authors consider to explore other metrics for measure the density, such as MMD or KL divergence?

---

> ### Public Comment · ~Georgi_Ganev1 · 2025-11-13
>
> Dear Reviewer `xwiG`,
>
> Thank you for raising the important point about Distance to Closest Record (DCR) being an insufficient measure of privacy leakage -- I fully agree with your observation.
>
> If I may suggest, the original work that first demonstrated this limitation is our paper [1], which is also cited in (Kaabachi et al., 2025) as the source of this finding. I would appreciate it if you could refer to it directly, as this might help clarify the origin of the result.
>
> [1] Ganev, Georgi, and Emiliano De Cristofaro. ``The Inadequacy of Similarity-Based Privacy Metrics: Privacy Attacks Against `Truly Anonymous' Synthetic Datasets.'' In IEEE S&P, 2025.

---

### Official Review · Reviewer_bu7E · 2025-11-03

**Soundness:** 2
**Presentation:** 3
**Contribution:** 3
**Rating:** 2
**Confidence:** 4

**Summary:**

The paper proposes TabSCM, a synthetic tabular data generator that explicitly models causal structure via structural causal models (SCMs). Starting from a CPDAG learned by any discovery algorithm, the method (i) orients edges to a DAG, (ii) fits marginals for root nodes (KDE for continuous, frequencies for categorical), and (iii) learns per-node structural assignments in topological order using conditional diffusion models for continuous children and gradient-boosted trees for categorical children. The design enables ancestral sampling, validity checks, and counterfactual interventions. Experiments on seven datasets across multiple domains compare TabSCM to GAN-, diffusion-, and LLM-based baselines.

**Strengths:**

1. This paper proposes to utilize structural causal models to generate date following the topological order and this method naturally support counterfactual reasoning.
2. The proposed method combines conditional diffusion with boosted trees to handle mixed type values.
3. The proposed method show competitive empirical results.

**Weaknesses:**

1. The proposed method is heavily depending on the causal discovery results. There lacks theoretical guarantee about how robust or in which condition the proposed method deals with the mis-specified structures, beyond the ablation study.
2. Identifiability is the central concern in causal inference and discovery. This work lacks investigation about the identification of the data generation and causal reasoning. It is challenging to get one-for-all solution but it is important to specify in which condition this method is identifiable.

**Questions:**

NA

---

### Meta-Review · Area_Chair_VkWh · 2026-01-06

**Summary:**

This paper develops a generative model for tabular data that explicitly represents causal structures encoded by structural causal models (SCMs). The proposed method combines diffusion models with boosting trees to construct a framework capable of achieving multiple objectives. The experimental results demonstrate certain advantages over several existing methods.

Reviewer bu7E argued that it is necessary to theoretically characterize which structures are learnable by the proposed approach and to provide a theoretical analysis of identifiability.

Reviewer xwiG acknowledged the practical advantages demonstrated in the experiments but argued that, due to identifiability issues, the main claims of the paper do not hold. The reviewer also pointed out that the related work and experimental comparisons are insufficient, noting in particular that several strong existing methods were overlooked. As a result, the advantages of the proposed method are not yet clearly established by the experiments. In addition, the reviewer criticized the lack of measurement of preprocessing time in the experimental setup and highlighted several arbitrary design choices. Overall, the reviewer provided a comprehensive critique, arguing that the improvements made in the study are inadequate.

Reviewer BjkV pointed out that the non-identifiability of the DAG structure remains unresolved and detailed multiple technical issues that have not been addressed. The reviewer further noted that handling counterfactual scenarios is not made technically explicit. Concerns regarding insufficient novelty and inadequate comparative evaluation were also raised in detail.

Overall, while the experimental results are presented with some clarity, multiple reviewers emphasized that the comparative evaluation is insufficient. More importantly, they highlighted that fundamental technical issues remain unresolved, and that the paper lacks comprehensive comparison with recent work as well as clear differentiation from existing methods.

The authors did not provide a rebuttal to address these concerns, and as a result, the identified issues remain unresolved.

**Reviewer Concerns:**

See above.

**Reviewer Scores:**

See above.

---

### Decision · Program_Chairs · 2026-01-26

Reject